# The Acidic Brain—Glycolytic Switch in the Microenvironment of Malignant Glioma

**DOI:** 10.3390/ijms22115518

**Published:** 2021-05-24

**Authors:** Anna Maria Reuss, Dominik Groos, Michael Buchfelder, Nicolai Savaskan

**Affiliations:** 1Laboratory for Translational Cell Biology and Neurooncology, Department of Neurosurgery, University Hospital Erlangen-Nürnberg, Friedrich-Alexander University Erlangen-Nürnberg (FAU), 91054 Erlangen, Germany; Michael.Buchfelder@uk-erlangen.de; 2Institute of Physiology and Pathophysiology, Friedrich-Alexander University Erlangen-Nürnberg (FAU), 91054 Erlangen, Germany; groos@hifo.uzh.ch

**Keywords:** tumor microenvironment, glycolytic, acidic, glioma, lactate, MCT1, MCT4, carbonic anhydrase (CA)IX, HIF, angiogenesis

## Abstract

Malignant glioma represents a fatal disease with a poor prognosis and development of resistance mechanisms against conventional therapeutic approaches. The distinct tumor zones of this heterogeneous neoplasm develop their own microenvironment, in which subpopulations of cancer cells communicate. Adaptation to hypoxia in the center of the expanding tumor mass leads to the glycolytic and angiogenic switch, accompanied by upregulation of different glycolytic enzymes, transporters, and other metabolites. These processes render the tumor microenvironment more acidic, remodel the extracellular matrix, and create energy gradients for the metabolic communication between different cancer cells in distinct tumor zones. Escape mechanisms from hypoxia-induced cell death and energy deprivation are the result. The functional consequences are more aggressive and malignant behavior with enhanced proliferation and survival, migration and invasiveness, and the induction of angiogenesis. In this review, we go from the biochemical principles of aerobic and anaerobic glycolysis over the glycolytic switch, regulated by the key transcription factor hypoxia-inducible factor (HIF)-1α, to other important metabolic players like the monocarboxylate transporters (MCTs)1 and 4. We discuss the metabolic symbiosis model via lactate shuttling in the acidic tumor microenvironment and highlight the functional consequences of the glycolytic switch on glioma malignancy. Furthermore, we illustrate regulation by micro ribonucleic acids (miRNAs) and the connection between *isocitrate dehydrogenase (IDH)* mutation status and glycolytic metabolism. Finally, we give an outlook about the diagnostic and therapeutic implications of the glycolytic switch and the relation to tumor immunity in malignant glioma.

## 1. Introduction

Malignant gliomas are the most common primary brain tumors, with an increasing incidence of up to nine per 100,000 inhabitants over the last years [1,2]. The highest prevalence is found in adults over 45 years of age. However, this extremely aggressive neoplasm can also affect younger people [3]. Malignant gliomas proliferate rapidly and diffusely infiltrate the surrounding brain tissue. Therefore, recurrence rates are high despite advances in surgical techniques and combined treatment with radio-chemotherapy [4,5]. In general, the prognosis is very poor, with a median survival of 12–15 months after diagnosis [6].

Malignant gliomas are classified according to the world health organization (WHO) grading system [7]. Histological criteria are mitotic activity for anaplastic astrocytoma (WHO grade 3) and microvascular proliferation and/or necrosis for grade 4. Of note, genetic alterations, such as *IDH* mutation status, have been shown to correlate more closely with the prognosis of malignant glioma than histological criteria alone. In fact, *IDH* mutant astrocytoma with necrosis and/or microvascular proliferation is classified separately from *IDH* wild-type (wt) glioblastoma, and designated *IDH* mutant astrocytoma, grade 4 [7,8].

Malignant gliomas are histologically heterogeneous with different tumor zones. Typically, glioblastoma multiforme (GBM) is characterized by central necrosis accompanied by microvascular proliferation leading to the typical ring contrast enhancement in magnetic resonance imaging (MRI). While the tumor cells at the leading edge of malignant glioma receive sufficient oxygen and nutrient supply from nearby blood vessels, the tumor cells in the perinecrotic center are under hypoxic conditions [9]. For a long time, this has been considered to represent a survival disadvantage for the tumor. However, mounting evidence shows that distinct tumor zones harbor a specific tumor microenvironment containing subpopulations of communicating cancer cells. A concomitant metabolic switch has been proposed to render cancer cells even more malignant and aggressive, leading to chemoresistance and tumor recurrence [10,11].

This review focuses on the glycolytic switch, causing an acidic tumor microenvironment and angiogenic switch in the hypoxic tumor center of malignant glioma. Special attention is paid to the role of lactic acid metabolism and MCTs, and the functional consequences on glioma malignancy. Furthermore, we highlight the regulation of glycolytic metabolism by miRNAs and the connection with *IDH* mutation status. Finally, we give an outlook about diagnostic and therapeutic implications of the glycolytic tumor microenvironment in malignant glioma and the relation to tumor immunity and immunotherapy.

## 2. Lactic Acid Metabolism within the Tumor Microenvironment

### 2.1. Lactic Acid Production—A Hallmark of Glycolytic Cancer Cells

Lactic acid is the end product of anaerobic glycolysis occurring mainly under hypoxic conditions and glucose deprivation [12,13,14]. Low partial pressure oxygen (pO_2_) leads to a glycolytic switch, i.e., the uncoupling of glycolysis from the tricarboxylic acid (TCA) cycle and oxidative phosphorylation (OXPHOS). Instead of entering the TCA cycle, pyruvate is converted directly to lactate by lactate dehydrogenase (LDH) (Figure 1A). The glycolytic switch is mediated particularly by HIF-1α. The reduction of negative feedback mechanisms by metabolites of glycolysis like glucose-6-phosphate (G6P), citrate, and adenosine triphosphate (ATP) is known as the “Pasteur effect”. Under normoxic conditions, the cytosolic protein HIF-1α is hydroxylated by prolyl hydroxylases (PHDs), which function as oxygen sensors with a very low affinity for oxygen (Michaelis *K*m value slightly above atmospheric concentration) [15,16]. Posttranslational modification targets HIF-1α to the E3 ubiquitin ligase von Hippel–Lindau (VHL) protein complex, where it is poly-ubiquitinated for proteasomal degradation [17,18] (Figure 1B). Under hypoxic conditions, oxygen sensor PHDs are inactivated, leading to the release of HIF-1α into the nucleus, where it binds to HIF-1β and further interacts with its cofactor protein (p)300/CREB binding protein (CBP) [19]. This complex binds to hypoxia-response elements (HREs) to initiate transcription of multiple genes, including those encoding glucose transporters (GLUTs), glycolytic enzymes, and enzymes that specifically drive anaerobic glycolysis [20,21,22,23] (Figure 1C).

Since the energy yield of anaerobic glycolysis is much lower than via the TCA cycle and OXPHOS, with only 2 molecules of ATP per molecule of glucose compared to 38 ATP, diverse oxygen, nutrient, and energy-sensing systems are activated to enhance the glycolytic flux through increased expression of glycolytic enzymes and transporters. The major advantage of anaerobic glycolysis is faster energy generation compared to ATP production via OXPHOS [24]. Therefore, highly glycolytic tissues, such as white skeletal muscle or tumors, show extensive lactate production even in the presence of oxygen. This phenomenon has been designated the “Warburg effect”, discovered almost 100 years ago [25]. Since this mechanism fulfills the high-energy demands of rapidly proliferating cancer cells, it has been suggested to sustain the proliferation of cancer cells by giving rise to biosynthetic pathways [14,25,26,27].

Therefore, the glycolytic phenotype, in which cancer cells regulate their energy consumption by switching from glucose to lactate, seems to enhance cancer cell survival and proliferation both under aerobic and hypoxic conditions.

In glioma cells, proteome analysis has confirmed a metabolic switch in response to hypoxia by the upregulation of GLUTs and all glycolytic pathway enzymes involved in lactate synthesis [28]. Furthermore, protein expression profiles revealed an aggressive epithelial-mesenchymal transition (EMT) and cancer stem cell (CSC) renewal phenotype of glioma cells under hypoxia, highlighting malignant transformation. Similarly, serum lactate levels have been proposed as a biomarker for glioma malignancy grade, showing significantly higher levels in high-grade versus low-grade gliomas [29].

### 2.2. The Role of MCTs in Lactic Acid Metabolism

#### 2.2.1. Structure and Function of MCTs

The MCT (*solute carrier* (*SLC16*)) family comprises 14 members with conserved sequence motifs and a common protein structure within the plasma membrane [30,31]. Multiple sequence alignments have shown identities ranging from 20% to 55%, with the highest conservation within the transmembrane domains (TMDs) and more variation in the C- than the N-terminal half. Two highly conserved sequence motifs at the start of TMD1 and TMD5 define the MCT family. Many conserved residues within the TMDs are glycines, which are likely important for forming turns, the packaging of helices, and the provision of flexible conformational changes. The conserved proline and other hydrophobic residues are considered to have structural significance. In contrast, the conserved charged and hydrophilic residues appear to have a catalytic role [30,31,32]. All MCTs have been predicted to contain 12 hydrophobic helical TMDs with an intracellular hydrophilic loop between TMD6 and TMD7 (Figure 2), ranging between 29 residues for MCT4 and 105 for MCT5. This loop divides the whole molecule into two halves with different functional roles. Whereas the N-terminal domains have been proposed to be important for H^+^/Na^+^ energy coupling, membrane insertion, and correct structure maintenance, the C-terminal domains have been suggested to determine substrate specificity. However, only the isoforms 1–4 of the mammalian MCT family, also showing the highest sequence conservation (>50%), have been demonstrated to function as real “monocarboxylate” transporters. MCTs1-4 act as H^+^ symporters via a suggested rapid equilibrium ordered mechanism reflected by H^+^ binding followed by monocarboxylate binding. The major substrate transported by MCTs1-4 in symport with H^+^ is L-lactate as the end product of anaerobic glycolysis. With a pKa of 3.86, lactic acid is almost entirely dissociated into lactate anions and protons within biological fluids [31,33]. In principle, the direction of transport is determined only by substrate and pH gradients across the plasma membrane. Thus all MCTs1-4 should be able to mediate influx or efflux of monocarboxylates. However, depending on substrate affinity reflected by the Michaelis *K*m value (Table 1), the high-affinity transporters MCT1 and MCT2 have been proposed to take up lactate and other monocarboxylates in low concentrations for further oxidation. In contrast, the low-affinity transporters MCT3 and MCT4 export lactate from highly glycolytic cells [34].

For proper translocation to and correct functioning within the plasma membrane, MCTs1-4 require permanent association with a glycosylated ancillary protein, consisting of a single TMD with a conserved glutamate residue, a short intracellular C-terminus, and two to three largely glycosylated extracellular immunoglobulin (Ig) domains depending on the splice variant [35,36] (Figure 2). Interestingly, in contrast to these highly glycosylated chaperones, none of the MCTs has been identified to be glycosylated itself for regulation. MCT1, MCT3, and MCT4 have been shown to form dimers, particularly with basigin (also known as cluster of differentiation (CD)147 or extracellular matrix metalloproteinase inducer (EMMPRIN)), whereas MCT2 prefers embigin/glycoprotein (GP)-70. However, dimer partners are promiscuous, differing between tissues and species dependent on the expression of the chaperone, which is for basigin more widely spread than for embigin [37]. CD2/basigin chimera experiments have revealed that the TMD and/or the intracellular tail of basigin rather than the extracellular domains are crucial for ancillary function [36]. Subsequent site-directed mutagenesis and molecular modeling analyses have led to the assumption that the single TMD of the chaperone interacts with cysteine residues on the external surface of TMD3 and TMD6 of MCTs [37,38,39]. The exploration of this association has led to a new possibility to specifically block MCT functioning instead of competitive inhibition. In this context, organomercurial agents, such as p-chloro-mercuri-benzene sulfonate (pCMBS), have been shown to bind to a labile disulfide bridge in the distal fold of the Ig-like C2 domain in basigin that is replaced by an unreactive Ig-like V2 domain in embigin [37,40]. The subsequent conformational change weakens its interaction with the bound MCT, thereby inhibiting transporter activity.

#### 2.2.2. An acidic Tumor Microenvironment and the Metabolic Symbiosis Model

*MCT1 (SLC16A1)* and *MCT4 (SLC16A3)* are overexpressed in patients with GBM compared to non-neoplastic control tissue and WHO grade 3 anaplastic and WHO grade 2 diffuse astrocytoma [41,42,43]. This overexpression is associated with poorer overall survival of GBM patients. Interestingly, *MCT1* is highly expressed at the leading tumor edge together with the *Na^+^/H^+^ exchanger (NHE**)1*, whereas *MCT4* is upregulated in the perinecrotic tumor center together with *HIF-1α, LDH*, and *CA**IX* in GBM patients and in a *rat* glioma model [43,44,45]. Consistent with this finding, *MCT4* expression is induced under hypoxic conditions in glioma cells, leading to enhanced lactic acid export and thereby decreased extracellular pH [43,46]. Accordingly, *MCT4* transcription is activated by HIF-1α, binding to two HREs within the *MCT4* promotor [47]. Likewise, *CAIX* expression is induced under hypoxic conditions via HIF-1α. It has been suggested to facilitate MCT1/4 activity by binding to the Ig domain of the MCT1/4 chaperone basigin in breast cancer cells [48,49]. Thus, glycolytic cancer cells upregulate MCT4 to export the produced lactic acid since intracellular acidification lowers the rate of glycolysis by inhibition of PFK and is further toxic for the cells. Indeed, the extracellular pH of most tumors has been determined to be much lower than in normal tissues [50]. For instance, electrode measurements of pH in *human* brain tumors have revealed a minimum of 5.9 and a mean of 6.8 compared to 7.1 in normal brain tissue. Furthermore, extracellular pH can vary considerably between distinct tumor regions. In vivo pH measurements in glioma using chemical exchange saturation transfer (CEST)-MRI has revealed that lower pH tumor regions are associated with higher proliferation index and expression of NHE1 and CAIX in subsequent immunohistochemistry [51]. Vice versa, lactate levels are already increased in anaplastic astrocytoma, and thus preceding the angiogenic switch leading to microvascular proliferation [52]. Multi voxel magnetic resonance spectroscopic imaging (MRSI) has detected maximum lactate concentrations in the center of GBMs [53]. Interestingly, extracellular acidosis in malignant glioma has been related to increased tumor survival and represents a crucial determinant for response to radio-chemotherapies [10,54].

In contrast to MCT4, the regulation of MCT1 is less clear. It is upregulated in glioma cells under hypoxia [46,55], but it does not possess HREs activated by HIF-1α under hypoxic conditions [47]. In addition, its *K*m value does not imply to function as a lactate exporter [56]. However, MCT1 has been suggested to function as a lactate exporter in glioma cells besides MCT4 [55].

In the metabolic symbiosis model, it has been proposed that tumor cells from distinct tumor zones communicate metabolically via the tumor microenvironment to sustain energy supply and fuel proliferation and survival of the entire tumor mass [9]. The model suggests that hypoxic tumor cells produce ATP via anaerobic glycolysis and upregulate MCT4 to release lactate into the tumor microenvironment, which fuels further lactate production by anaerobic glycolysis. Via MCT1, lactate is then preferentially taken up by oxygenated tumor cells adjacent to blood vessels because lactate in contrast to glucose spares energy normally spent for housekeeping glycolytic enzymes. Lactate is converted to pyruvate to fuel the TCA cycle and OXPHOS under aerobic conditions. This mechanism enhances the glucose gradient, ensuring energy delivery into hypoxic tumor regions, and therefore, the overall survival of the tumor. This model is consistent with findings that *MCT4* is upregulated in the perinecrotic core of GBM, whereas *MCT1* is mainly overexpressed at the leading edge [43,44]. Furthermore, it has been shown that under glucose deprivation, lactate preserves high ATP levels in glioma cells [44], sustaining the high-energy demands of proliferating tumor cells. Interestingly, there is evidence that both lactate and pyruvate can activate hypoxia-sensitive genes independent from hypoxia by promoting HIF-1α accumulation [57]. This finding indicates that glycolytic end products may also lead to a hyper-glycolytic phenotype under normoxic conditions by a positive feedback mechanism. On the other hand, lactic acid has been reported to convert the dominant Warburg effect to OXPHOS at the tumor edge, where HIF-1α is decreased, whereas cellular-myelocytomatosis (c-MYC), nuclear respiratory factor (NRF)1 and OXPHOS related proteins are increased [44].

In a nutshell, these findings lead to the following working model in malignant glioma (Figure 3A).

### 2.3. Lactic Acid Metabolism through MCTs and Functional Consequences on Tumor Malignancy

The glycolytic switch from OXPHOS to lactic acid metabolism with upregulation of *MCT4* in the hypoxic center of malignant glioma, leading to a hyper-glycolytic phenotype and an acidic tumor microenvironment, has been suggested to render tumor cells more aggressive. Acidosis in the tumor microenvironment consistently correlates with tumor growth and survival, invasion, and chemoresistance [10,54,58]. Likewise, lactate serum and tumor tissue levels are associated with malignancy grade in glioma [29,59]. Functional studies in glioma cells and GBM neurospheres have shown that MCT4 increases tumor cell proliferation and survival, whereas MCT4 knockdown or pharmacological inhibition leads to apoptosis and necrosis [43,60]. Furthermore, MCT4 enhances migration and invasiveness by reorganization of the actin cytoskeleton [43]. Export of lactate and protons decreases extracellular pH and increases intracellular pH. Elevated intracellular pH has been demonstrated to remodel the cytoskeleton for migration and invasion [61,62]. Moreover, invadopodia-mediated extracellular matrix degradation, as well as activation of extracellular matrix dissolving enzymes, is augmented by an acidic tumor microenvironment [63,64]. Proteomic analysis has detected EMT in glioma cells exposed to hypoxia [28] and transcriptomic analysis in glioma transitioning to higher malignancy [65]. Accordingly, upregulation of *MCT4* is higher in GBM with mesenchymal molecular subtype than in other molecular subtypes [43,53]. Interestingly, hypoxia induces a CSC phenotype in glioma cells along with *MCT4* expression [28,53], which is depleted under *MCT4* silencing [60].

Similarly, MCT1 has been found to play an important role in glioma malignancy. *MCT1* is also associated with EMT and CSC phenotype, with selective MCT1 inhibition decreasing the viability of glioma stem cells [53,66]. MCT1 knockdown or inhibition via cyano-hydroxycinnamate (CHC) reduces glioma cell proliferation, migration, invasion, and glycolytic metabolism. Furthermore, it induces cancer cell death [55,67].

Like *MCT*s, their facilitator *CAIX* is overexpressed under hypoxia in the perinecrotic regions of GBM via epidermal growth factor receptor (EGFR) and signal transducer and activator of transcription (STAT)3 signaling, which correlates with reduced overall survival [48,68,69,70,71]. Interestingly, CAIX expression is already relevant for patient survival in grade 2 and 3 astrocytoma [68]. Functional studies have revealed that CAIX is associated with increased growth, migration, and invasion [70,71,72]. Interestingly, CAIX expression also correlates with enhanced necrosis and apoptosis and is inhibited by acidity, suggesting a self-limiting mechanism [72].

A summary of the functional consequences of the glycolytic switch on glioma malignancy is depicted in Figure 3 (Figure 3B).

### 2.4. Lactic Acid Shuttling and the Induction of Angiogenesis

Besides the glycolytic switch, the angiogenic switch is a hallmark of malignant glioma. While the tumor cells at the leading edge grow invasively towards adjacent blood vessels in the surrounding normal brain tissue, hypoxic tumor cells in the center promote angiogenesis, leading to neovascularization [73]. Lactate has been proposed to stimulate angiogenesis through HIF-1α dependent activation of the vascular endothelial growth factor (VEGF) signaling pathway [74]. Consistently, low extracellular pH has been related to the induction of *VEGF* via the mitogen-activated protein kinase (MAPK) pathway in *human* glioma cells in vitro as well as in brain tumors In vivo [75,76].

Similar to the metabolic symbiosis model between oxidized and glycolytic tumor cells, interaction with nearby blood vessels via lactate shuttling has been suggested [77,78]. Lactate released by glycolytic tumor cells via MCT4 is taken up by endothelial cells via MCT1, thereby supporting pro-angiogenic signaling via HIF-1α and an autocrine nuclear factor (NF)κB/interleukin (IL)8 pathway (Figure 3C). Consistently, *MCT4* overexpression is found in the hyperplastic/microvascular proliferation zones in GBM associated with *VEGF* upregulation in endothelial cells, whereas MCT4 inhibition reduces the induction of angiogenesis [43]. Likewise, exposure of *human* brain microvascular endothelial cells (HBMEC) to glioma cell-conditioned media upregulates *MCT1* expression and stimulates angiogenesis [79]. Furthermore, lactate treatment of HBMEC activates protein kinase B (AKT), and adenosine monophosphate protein kinase (AMPK) signaling pathways and increases expression of *HIF-1α, NFκB*, and the lactate *G-protein receptor (GPR**)81*. Recently, it has been proposed that hypoxic glioma cells upregulating MCT1 and basigin release more exosomes containing these metabolic markers in a calcium-dependent manner, which are taken up by endothelial cells where they lead to the induction of angiogenesis [80].

Moreover, CAIX overexpression in glioma has been related to increased VEGF levels and enhanced microvascular density and microvessel caliber parameters, which is associated with shorter patient survival [68].

Like hypoxia and MCT4, the angiogenic switch has been linked to higher glycolysis rates and EMT, with most EMT-associated genes found in the hyperplastic/microvascular proliferation tumor zones [43,81]. Interestingly, the angiogenic and mesenchymal switch coincides with *STAT3* upregulation, regulating *HIF-1α* and *CAIX* expression under hypoxia in glioma cells [71].

In summary, these findings underscore that the glycolytic and angiogenic switch are highly intertwined, fostering malignancy of glioma cells via the tumor microenvironment (Figure 3).

## 3. The Role of miRNAs in the Glycolytic Switch

Compared to healthy brain tissue, a multitude of miRNAs is dysregulated in glioma. While miRNAs acting as tumor suppressors are downregulated, those acting like oncogenes are upregulated. Several studies have identified downstream miRNA target genes as well as functional consequences for tumor malignancy both in vitro and In vivo (Table 2). The limitation of cellular glucose uptake or lactate secretion and thereby preventing a glycolytic switch is achieved by several miRNAs via direct and indirect targeting of respective genes. For instance, under physiological conditions, *miR-495* inhibits glucose uptake by directly suppressing *GLUT1 (SLC2A1)* [82]. In glioma, downregulation of *miR-495* prevents *GLUT1* suppression, leading to increased glucose uptake, lactate secretion, and cell proliferation. Another important regulator of glucose uptake is *miR-451*, whose expression is regulated by a glucose level mediated feedback mechanism. While *miR-451* is abundantly expressed under high glucose levels, low glucose levels for as long as 24 h are sufficient to downregulate *miR-451* [83,84]. Consequently, disinhibition of downstream genes, such as *CAB39*, leads to augmented glucose uptake and lactate secretion associated with enhanced proliferation, viability, migration, and invasion of glioma cells [83,84,85,86,87]. On the other end, there are miRNAs, which are specifically overexpressed in glioma. For example, upregulation of *miR-150*, targeting the tumor suppressor *VHL*, increases HIF-1α expression levels. In turn, HIF-1α promotes glucose uptake, glycolysis, and lactate secretion through the upregulation of *GLUT1* and glycolytic enzymes, thereby fostering cell proliferation and tumor growth [88]. Taken together, miRNA dysregulation in glioma disables proper tumor suppression, increases glycolytic metabolism, and augments tumor malignancy through multiple effectors and signaling pathways. A detailed overview is depicted in Table 2 (Table 2).

## 4. The Acidic Tumor Microenvironment in *IDH* wt Versus *IDH* Mutant Gliomas

In recent years, the major prognostic relevance of *IDH* mutation status has been recognized, heralding the era of the integrated histological-molecular approach for diagnosing brain tumors [7,8]. Missense mutations in the *IDH1* or *IDH2* gene lead to the replacement of positively charged arginine residues by non-charged, polar amino acids, such as histidine or cysteine [108,109]. The resultant impairment of hydrogen bond formation with the carboxy sites reduces the affinity for isocitrate. It increases the preference for nicotinamide adenine dinucleotide phosphate (NADPH) [110]. Since most cancer cells harbor heterozygous *IDH* mutations, *IDH* heterodimers reveal, on one hand, the wt form, converting isocitrate and NADP^+^ into α-ketoglutarate (KG), carbon dioxide (CO_2)_, and NADPH. On the other hand, the mutant form displays neomorphic activity, converting α-KG into the enantiomer D-2-hydroxyglutarate (D-2-HG) in an NADPH-dependent manner (Figure 4).

D-2-HG inhibits α-KG dependent enzymes and leads to a hypermethylation phenotype [111,112,113,114]. Since *IDH* is a crucial enzyme in the TCA cycle, other metabolic pathways in the tumor microenvironment may be impacted by epigenetic regulation.

Initially, *IDH1* mutated glioma cells have been reported to produce lower amounts of α-KG in favor of D-2-HG, leading to HIF-1α overexpression. Thus, IDH1 has been suggested to act as a tumor suppressor, which, when inactivated by mutation, contributes to tumorigenesis in part by induction of the HIF-1α pathway [115].

In contrast, another study has shown that D-2-HG stimulates PHDs, leading to reduced HIF-1α levels in *IDH* mutant cells [116]. Simultaneous pH and oxygen-sensitive MRI have revealed higher acidity and higher hypoxia in *IDH* wt than *IDH* mutant gliomas, correlating to higher HIF-1α expression and increased proliferation index [117]. Accordingly, HIF-1α target genes, including many glycolytic genes, are downregulated in *IDH* mutant compared to *IDH* wt gliomas [118]. Further gene expression analysis has revealed a glycolytic phenotype for *IDH* wt gliomas. In contrast, *IDH* mutant gliomas overexpress genes encoding for TCA cycle involved enzymes [119]. This finding has been confirmed by in vitro experiments, showing that *IDH* mutant HCT 116 cells have higher basal respiration rates than *IDH* wt cells. Likewise, lactate levels are significantly increased in *IDH* wt gliomas. In contrast, *IDH* mutant cells exhibit reduced hyperpolarized lactate production along with decrease of LDH as well as MCT1 and MCT4 levels [120,121,122,123].

Therefore, epigenetic silencing of glycolytic switch-related genes may explain why *IDH* mutant gliomas exhibit slower proliferation and less aggressive behavior than *IDH* wt gliomas [122].

On the other hand, acquisition of the Warburg phenotype has been associated with the CpG island methylator phenotype (G-CIMP) in *IDH* mutant astrocytoma, displaying more aggressive behavior than *IDH* mutant oligodendroglioma [124]. Consistently, D-2-HG overproduction inhibits OXPHOS in *IDH1^R132H^* mutant glioma cells, thereby reducing ATP levels and activating the AMPK pathway [125]. AMPK-mediated inhibition of the mammalian target of rapamycin (mTOR) signaling decreases protein synthesis, making *IDH* mutant glioma cells vulnerable to synthetic lethality through inhibition of B cell lymphoma-extra large (Bcl-xL). This finding is consistent with a prior study, showing that D-2-HG inhibits ATP synthase and that *IDH1^R132H^* mutant cells display reduced ATP levels and mTOR signaling [126].

However, another study has suggested that D-2-HG leads to mTOR activation in *IDH1^R132H^* mutant cells by disinhibition via the lysine demethylase (KDM)4A [127].

The PIK3/AKT/mTOR pathway is known to enhance glucose uptake via GLUT1 and to control the glycolytic flux through regulation of glycolytic enzymes without affecting the rate of OXPHOS [128,129,130]. Under hypoxic conditions, mTOR has been suggested to activate HIF-1α, thereby promoting the glycolytic switch [131].

Regarding these intricate signaling pathways, it is conceivable that *IDH* mutation can regulate glycolysis connected to OXPHOS in glioma cells. When undergoing the Warburg effect or under hypoxic conditions, *IDH* mutant cells may also promote the glycolytic switch. Therefore, the effects are likely highly dependent on the cell status and predominant microenvironment in *IDH* mutant gliomas, which may vary considerably between different grades. However, the relation between *IDH* mutation, D-2-HG, and HIF-1α is still controversial and needs further clarification.

It is clear that *IDH* mutations, redirecting carbon metabolites away from the TCA cycle towards D-2-HG production, decrease oxidative metabolism and shift the redox potential to a more oxidated state. The resultant increase in oxidative stress has been related to the enhanced sensitivity of *IDH* mutant cells to chemotherapy [132,133,134,135,136].

## 5. Implications for Diagnostics and Therapy

Malignant glioma is a fatal neoplasm with a very poor prognosis despite advances in surgical techniques and combined treatment with radio-chemotherapy [4,5,6]. In this review, it has been highlighted that gliomas with glycolytic and angiogenic switch tumor microenvironments display more aggressive and malignant behavior. Therefore, identification of these specific tumor regions In vivo and local, specific targeting of glycolytic glioma cells may give rise to new therapeutic approaches.

### 5.1. Diagnostic Approaches for Identifying Glycolytic Tumor Regions

Hypoxic brain regions with low pH and elevated lactate levels can be identified in patients using MRI and positron emission tomography (PET) techniques [51,53,137,138]. Radiotracers for MCT and CAIX imaging in brain gliomas are also under development [139,140]. Furthermore, MRSI has been indicated for the monitoring of metabolic responses to treatments via the lactate-to-pyruvate-ratio, which is more sensitive than evaluation of tumor growth by conventional MRI [122]. Interestingly, MRSI monitoring of the lactate-to-pyruvate-ratio during treatment of glioma cells with the histone deacetylase (HDAC) inhibitor vorinostat has revealed a significant decrease, accompanied by upregulation of MCT4 and MCT1 [141]. Therefore, MRSI may serve as an important monitoring tool to detect resistance mechanisms during the therapy of malignant glioma.

### 5.2. Glycolytic Players as Targets in Glioma Therapy

As mentioned above, inhibitor treatment of MCTs in glioma cells significantly reduces proliferation and survival, migration and invasiveness, and the induction of angiogenesis [43,53,55,60,66,67]. General MCT inhibition may be problematic due to physiological expression in skeletal muscle and brain astrocytes, where astrocyte-neuron lactate shuttling is crucial for memory formation [30,142,143]. However, programmed orthotopic administration of the MCT inhibitor CHC by osmotic pumps into gliomas implanted into *rats* has been shown to substantially decrease invasion and to lead to necrosis within the tumor bed [144]. Importantly, no neurological side effects have been observed. Likewise, the small molecule acriflavine, targeting the binding between MCT4 and its chaperone basigin, has been revealed to inhibit the growth and self-renewal potential of glioma neurospheres, especially under hypoxia [145]. In stem cell-derived xenograft *mice*, acriflavine markedly reduces tumor progression and vascularization by VEGF inhibition. Vice versa, hypoxic and glycolytic markers like HIF-1α, CAIX, GLUT1, and MCT1 are upregulated in glioma cells under bevacizumab treatment [146]. This may explain the resistance that gliomas acquire during therapy. In contrast, inhibition of CAIX or glucose uptake enhances the effects of bevacizumab [72,146]. Likewise, pretreatment of glioma cells targeting CAIX or MCT4 increases sensitivity to subsequent radio-chemotherapy [70,147]. Moreover, a combination of the conventional chemotherapeutic agent temozolomide with the CAIX inhibitor acetazolamide significantly enhances cell death of glioma cells and glioma stem cells under hypoxic conditions [148].

Furthermore, *HIF-1α* has been suggested as a potential therapeutic target using small interference (si)RNA, packaged in a novel surfactant-based nucleic acid carrier and delivered into gliomas of an In vivo orthotopic *mouse* model by osmotic pumps [149]. *HIF-1α* silencing is associated with the downregulation of the transcriptional targets *GLUT1*, *CAIX*, and *VEGF*. It has reduced the tumor volume by 79% after 50 days of daily treatment. Proliferation index and microvascular density have also been significantly lower.

Additionally, other glycolytic inhibitors like 3-bromopyruvate (3-BP), which is structurally related to lactate and pyruvate, the PFK inhibitor citrate and enolase inhibitor sodium fluoride, have been proposed as energy depletion therapy in glioma cells [150]. 3-BP induces caspase-dependent cell death and blocks migration of glioma cells promoted by lactate. Notably, 3-BP and citrate show synergistic effects in decreasing glioma cell viability.

Besides apoptosis and necrosis, ferroptosis is a recently proposed iron-dependent mechanism of cell death [151]. Interestingly, *MCT4* overexpressing glioma cells show sensitivity to ferroptosis compared to normal or reduced *MCT4* expression levels [43]. This finding indicates another selective therapeutic approach for locally targeting exclusively *MCT4* overexpressing cells in malignant glioma, which warrants further exploration.

Finally, metabolic changes in the hypoxia-induced tumor microenvironment under EGFR and mTOR inhibition, leading to adverse effects, have suggested the exact opposite approach [152]. Instead, mTOR activation by suppressing its physiological inhibitor *tuberose sclerosis complex (TSC)**2* causes hypoxia-induced glioma cell death by earlier ATP depletion and reactive oxygen species (ROS) production. Identification of the underlying mechanisms has revealed enhanced oxygen consumption due to the upregulation of genes involved in OXPHOS and increased metabolites of the pentose phosphate pathway due to upregulation of the rate-limiting enzyme G6P. This finding may also be beneficial in combination with hypoxia-inducing therapies like bevacizumab treatment.

In summary, exploiting the glycolytic switch for selective tumor therapy offers a broad range of novel possibilities, which malignant glioma patients may profit from soon.

### 5.3. Impact of the Glycolytic Phenotype on Tumor Immunity and Immunotherapy

The immune system’s role in tumorigenesis and its exploitation for immunotherapy has been an emerging field in recent years. For a general review about immunotherapy in glioma, we refer to [153].

In glioma, expression of immune checkpoint genes is associated with more aggressive tumor behavior and worse prognosis [154]. Interestingly, genomic analysis has revealed an epigenetic link between glycolytic and immune checkpoint gene expression in low-grade glioma [155]. In this study, the *IDH* wt cluster displayed lower levels of *LDHA* promoter methylation and a higher *LDHA/LDHB* expression ratio. This genotype was accompanied by less promotor methylation of the immune inhibitory molecule *programmed cell death ligand (PDL**)1/2* and thus higher *PDL1/2* expression levels. In contrast, *IDH^R132H^* induction decreased promotor histone (H)3K4 triple methylation (me3) for *LDHA* and *PDL1/2*. Crosstalk between the immune checkpoint and metabolic pathways may profoundly impact tumor cell evasion from immune system recognition. For instance, in glioma, uncoupling protein (UCP)2 has been proposed to link the glycolytic switch to dampened immune response [156].

Active immune cells also exhibit the Warburg effect to fulfill their energy demands. For example, quiescent naïve T lymphocytes use OXPHOS, whereas activation induces the glycolytic switch in these cells [157,158]. Therefore, nutrient competition may be one metabolic mechanism involved in tumor cell evasion from the immune system. Indeed, glycolytic tumors with overexpression of *GLUT1* and *LDHA* and enhanced lactate secretion show an inverse correlation with infiltration of CD8^+^ cytotoxic T lymphocytes (CTLs) [159]. Furthermore, glycolytic tumor-infiltrating T cells exhibit fewer effector molecules, such as granzyme B and perforin, and thus reduced cytotoxicity. In a *mouse* sarcoma model, it has been demonstrated that increased glycolysis and thus glucose consumption in tumor cells metabolically restricts T lymphocytes by reducing mTOR activity, glycolytic capacity, and interferon (IFN)-γ production in these cells [160]. Abrogation of proper T cell function is sufficient to promote tumor growth. In contrast, checkpoint blockade antibodies against CTL-associated protein (CTLA)4, PD1, and PDL1 restores glucose levels in the tumor microenvironment, T cell function, and IFN-γ production. Mechanistically, PDL1 blockade has been shown to decrease glycolysis in tumor cells by inhibiting mTOR activity and reduced the expression of glycolytic enzymes. However, rapamycin-mediated inhibition of immune checkpoint molecule induced mTOR pathway in naïve CD8^+^ T cells promotes the production of memory T cell precursors by persistent *eomesodermin* expression [161]. These cells show enhanced antigen-recall responses upon adoptive transfer and higher tumor efficacy. Consistently, it has been demonstrated that enhanced glycolysis in CTLs impairs memory T cell generation by driving them towards a terminally differentiated state [162]. Notably, these cells fail during adoptive transfer. In contrast, inhibition of glycolysis in tumor-specific CTLs increases antitumor response.

Vice versa, tumor cell glycolysis has been associated with immune resistance to adoptive T cell therapy in melanoma [163]. Highly glycolytic melanoma cells display reduced levels of *interferon regulatory factor (IRF**)1* and C-X-C motif chemokine ligand (CXCL)10 immunostimulatory molecules.

Furthermore, hypoxia leads to HIF-1α dependent *PDL1* upregulation in tumor cells, thereby increasing resistance to CTL lysis [164]. Hypoxia-mediated *PDL1* overexpression in tumor cells increases apoptosis in co-cultured T cells, which is abrogated by blocking the interaction of PDL1 and the PD1 receptor on T cells. Furthermore, treatment with glyceryl trinitrate (GTN), an agonist of nitric oxide (NO) signaling known to block HIF-1α accumulation in hypoxic cells, prevents hypoxia-mediated *PDL1* overexpression and reduces T cell apoptosis. This is accompanied by decreased tumor cell immune escape from CTL-mediated lysis and reduces tumor growth, suggesting novel cancer immunotherapy to block *PDL1* expression specifically in hypoxic tumor cells by administering NO mimetics.

Moreover, *PD1* intrinsically expressed in melanoma cell subpopulations has been shown to promote tumorigenesis via mTOR signaling, even in *mice* lacking adaptive immune response [165]. In contrast, PD1 inhibition by antibodies, siRNA or mutagenesis of *PD1* signaling motifs substantially reduces tumor growth.

Interestingly, it has been shown that *PD1* overexpression in tumor-infiltrating T lymphocytes during prolonged antigen exposure leads to DNA methylation and is responsible for complete T cell exhaustion, which is resistant to immune checkpoint blockade mediated rejuvenation [166]. Intriguingly, the authors propose approaches for DNA methylation reprogramming, which improves T cell responses and tumor control during immune checkpoint blockade. This discovery may also give rise to developing novel strategies in glioma immunotherapy.

Besides suppressing tumor immunity via immune checkpoint modulation in glycolytic tumor cells, a direct inhibitory effect of tumor cell-derived lactic acid on CTL proliferation, cytokine production, and cytotoxicity against tumor spheroids has been shown [167]. The authors propose that high lactic acid levels within the tumor microenvironment block the export of lactic acid from T cells through MCT1, thereby disturbing their metabolism and function.

These results suggest that targeting glycolytic pathways in tumor and T cells combined with immunotherapy opens new perspectives in cancer treatment. For instance, CAIX has been demonstrated as a suitable target for selective chimeric antigen receptor (CAR) T cell therapy with a cure rate of 20% and without any systemic side effects in an In vivo glioma xenograft *mouse* model [168].

Besides T lymphocytes, tumor-associated macrophages (TAMs) play a crucial role in glioma immunity and glioma cell evasion [169]. Interestingly, glioma cells have been shown to secrete branched-chain ketoacids via MCT1, taken up by TAMs, where they reduce phagocytosis [170]. This finding raises the possibility of a role of MCTs in tumor immune suppression. Furthermore, upregulated CAIX leads to TAM M2 polarization, indicating an immunosuppressive phenotype in glioma [71]. M2 TAMs produce IL-1β, which in turn leads to phosphorylation of the glycolytic enzyme GA3PDH in glioma cells through PIK3-mediated activation of protein kinase (PK)Cδ [171]. Blocking of the respective factors reduces the glycolytic rate and proliferation of glioma cells. In patient-derived glioma tissue, staining for IL-1β and macrophages correlates with PKCδ and GAP3DH, and this is associated with higher glioma grade and lower overall survival.

Taken together, these findings suggest interference of cytokine crosstalk between M2 TAMs and glioma cells as a further possible treatment approach.

Finally, it has been shown that sensitization of glioma cells by low-dose administration of attenuated oncolytic measles virus Edmonston strain (MV-Edm), leading to a shift to high-rate aerobic glycolysis, to the glycolytic blocker dichloroacetate, induces substantial cell death of tumor, but not of non-tumor cells in vitro as well as in an In vivo GBM xenograft model [172]. Besides blocking bioenergetic generation, dichloroacetate enhances viral replication by abrogating mitochondrial antiviral signaling protein (MAVS)-mediated innate immune response, increasing bioenergetic consumption and oncolysis.

This somewhat uncommon approach seems to be quite powerful by combining different effects and should be considered for further investigation.

In summary, exploiting the glycolytic phenotype for combined immunotherapy with distinct components in tumors and different immune cells may constitute a promising therapeutic approach. Due to the combined targeting of various effectors, is likely more powerful and less prone to resistance mechanisms than focusing on single mechanisms. Large and sophisticated studies are necessary to explore the complexity of such an approach in detail.

## 6. Conclusions and Future Perspectives

This review has shown that malignant gliomas are heterogeneous neoplasms composed of distinct tumor zones whose subpopulations of cancer cells communicate via specific tumor microenvironments. Glioma cells in the perinecrotic tumor center live under hypoxic conditions. These cancer cells undergo a glycolytic switch from OXPHOS to anaerobic glycolysis via regulation by HIF-1α, inducing upregulation of glycolytic enzymes, transporters, and *VEGF*. This, in turn, triggers the angiogenic switch, leading to neovascularization. Lactate, the end product of anaerobic glycolysis, is exported as lactic acid primarily via MCT4, thereby increasing acidity in the tumor microenvironment. The low extracellular pH elevated lactate levels, and other metabolites produced by the high glycolytic flux lead to metabolic gradients across the tumor microenvironment, remodel the extracellular matrix and activate signaling pathways in neighboring cells. Of note, lactic acid uptake by endothelial cells via MCT1 leads to the induction of angiogenesis. Neovascularization, migration and invasion, as well as proliferation and survival, are the functional consequences, rendering malignant gliomas even more aggressive and resistant to established treatment regimens.

However, the glycolytic switch can also be exploited for developing novel therapeutic approaches by taking advantage of the weak points in glycolytic cells, such as sensitivity to oxidative stress and hypoxia-induced cell death. Modern therapeutic concepts or their combination with conventional treatment regimens or immunotherapy may overcome cancer cell resistance to achieve a better prognosis for malignant glioma patients. The development of selective targeting, applicable to glioma patients without severe side effects, especially for other glycolytic cells like skeletal muscle or brain astrocytes, represents the crucial challenge for future studies.

## Figures and Tables

**Figure 1 ijms-22-05518-f001:**
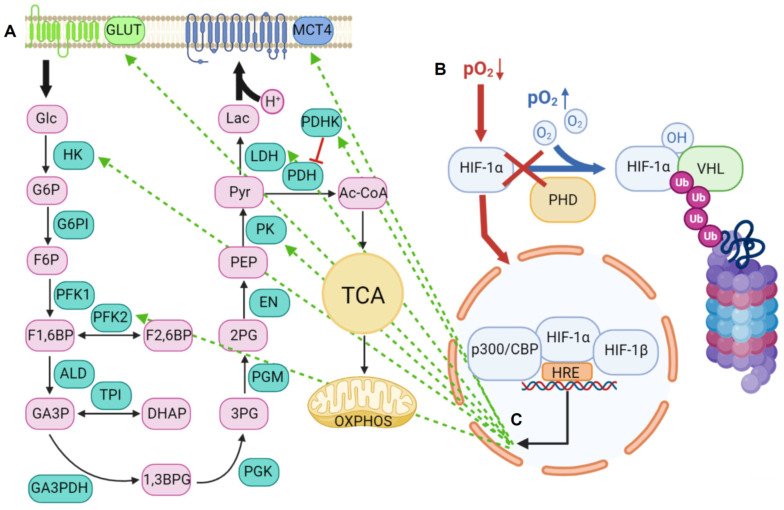
The glycolytic switch under hypoxia. (**A**) Scheme of glycolysis. Glucose (Glc) is taken up via GLUTs and enters glycolysis. The product pyruvate (Pyr) enters the TCA cycle and OXPHOS by conversion to acetyl coenzyme A (Ac-CoA) in oxidative cells or is converted to lactate (Lac) under anaerobic conditions in glycolytic cells. Lactic acid is exported by MCT4. (**B**) Regulation of HIF-1α. Under normoxia, HIF-1α is hydroxylated by PHDs and ubiquitinylated (Ub) by VHL complex for proteasomal degradation. Under hypoxia, HIF-1α migrates into the nucleus and activates transcription of multiple genes by binding to HREs in a complex with HIF-1β and p300/CBP. (**C**) Regulation of the glycolytic switch by HIF-1α. Under hypoxia, HIF-1α induces GLUTs, MCT4, and different glycolytic enzymes, which direct glucose consumption into anaerobic glycolysis. G6P: glucose-6-phosphate, F6P: fructose-6-phosphate, F1,6BP: fructose-1,6-bisphosphate, F2,6BP: fructose-2,6-bisphosphate, GA3P: glycerin-aldehyde-3-phosphate, DHAP: di-hydroxy-acetone–phosphate, 1,3BPG: 1,3-bis-phosphoglycerate, 3PG: 3-phospho-glycerate, 2PG: 2-phospho-glycerate, PEP: phospho-enol-pyruvate, HK: hexokinase, G6PI: glucose-6-phosphate-isomerase, PFK1/2: phospho-fructokinase 1/2, ALD: aldolase, TPI: triose-phosphate-isomerase, GA3PDH: glycerin-aldehyde-3-phosphate-dehydrogenase, PGK: phospho-glycerate-kinase, PGM: phospho-glycerate-mutase, EN: enolase, PK: pyruvate-kinase, PDH: pyruvate-dehydrogenase, PDHK: pyruvate-dehydrogenase-kinase.

**Figure 2 ijms-22-05518-f002:**
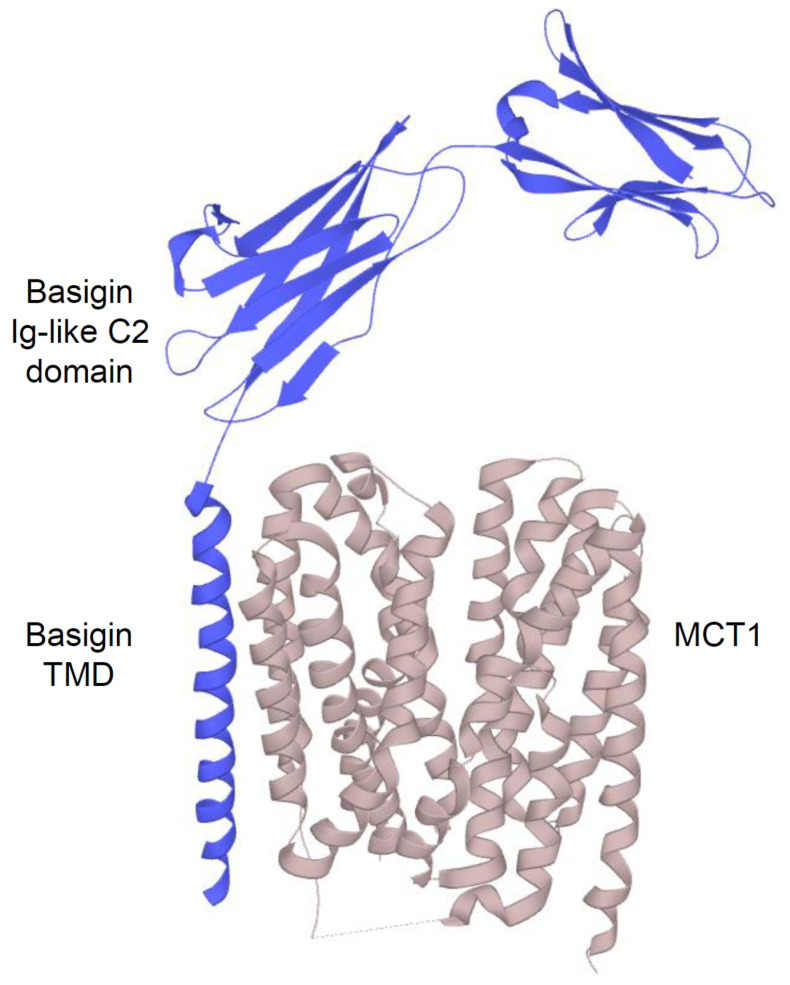
Cryo-electron microscopy (EM) structure of *human* MCT1/basigin complex. MCT1 with 12 hydrophobic helical TMDs (**beige**) forms a functional complex with its chaperone basigin (**blue**). Data were obtained from UniProt (P53985).

**Figure 3 ijms-22-05518-f003:**
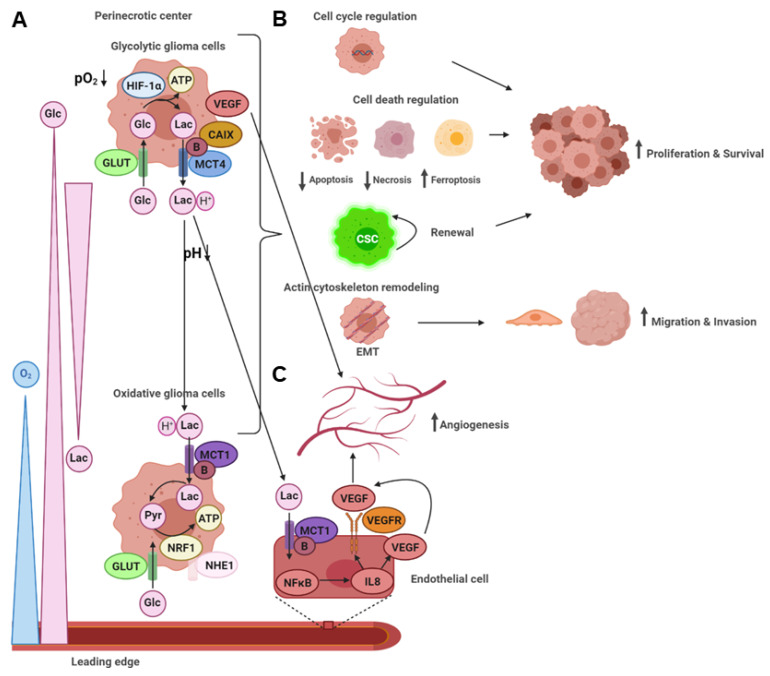
Glycolytic and angiogenic switch in the tumor microenvironment. (**A**) The metabolic symbiosis model between oxidative and glycolytic glioma cells. Under hypoxia, glioma cells in the perinecrotic center upregulate different genes regulated by HIF-1α. Oxidative glioma cells at the leading edge show a different gene expression profile. Glycolytic glioma cells release lactic acid into the tumor microenvironment, leading to acidity. Lactic acid is taken up by oxidative glioma cells, preferably to glucose, thereby enhancing the glucose gradient from blood vessels to glycolytic glioma cells in the tumor center. (**B**) Functional consequences on glioma malignancy. The glycolytic switch leads to increased proliferation and survival by cell cycle and cell death regulation, a CSC phenotype, and enhanced migration and invasion by EMT and actin cytoskeleton remodeling. (**C**) Glycolytic and angiogenic switch. The glycolytic switch induces angiogenesis by *VEGF* upregulation in glycolytic glioma cells and stimulation of autocrine VEGF signaling in endothelial cells by lactate. B: basigin.

**Figure 4 ijms-22-05518-f004:**
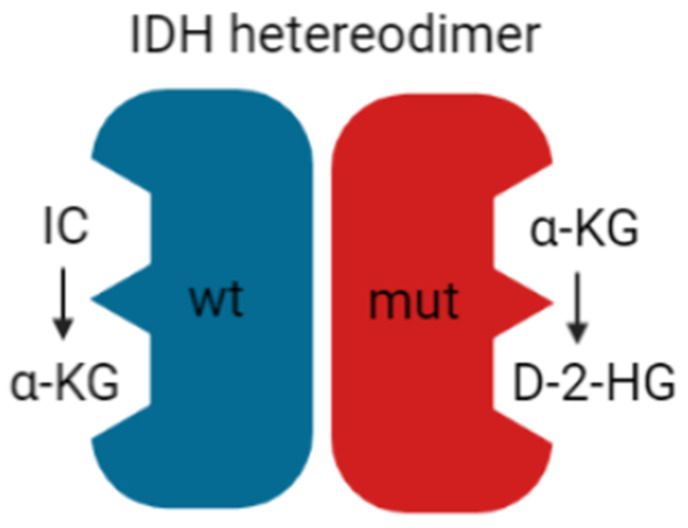
*IDH* heterodimer. *IDH* wild-type (wt; **blue**) and mutant (mut; **red**) monomers form a catalytically active heterodimer. *IDH* wt converts isocitrate (IC) to α-KG while *IDH* mut converts α-KG to D-2-HG.

**Table 1 ijms-22-05518-t001:** *K*m values for MCTs 1, 2 and 4. *K*m values in mM determined in *Xenopus* oocytes. Modified from Halestrap, 2012.

Substrate	MCT1	MCT2	MCT4
Formate	-	-	>500
Bicarbonate	-	-	>500
Oxamate	-	-	>500
Glyoxylate	-	-	>500
L-Lactate	3.5	0.74	28
D-Lactate	>60	-	519
Pyruvate	1.0	0.08	153
S-Chloropropionate	-	-	46
R-Chloropropionate	-	-	51
D,L-α-Hydroxybutyrate	-	-	56
L-β-Hydroxybutyrate	-	1.2	824
D-β-Hydroxybutyrate	-	1.2	130
γ-Hydroxybutyrate	-	-	>500
Acetoacetate	-	0.8	216
α-Ketobutyrate	-	-	57
α-Ketoisocaproate	0.7	0.1	95
α-Ketoisovalerate	1.3	0.3	113
β-Phenylpyruvate	-	-	>500

**Table 2 ijms-22-05518-t002:** Dysregulated miRNAs in glioma cells.

miRNA	Expression in Glioma	Targeted by	Targets	Effects in Glioma	Literature
*miR-1*	Downregulated	-	*Annexin A2*	Decreases proliferation, invasion, and angiogenesis in glioma cells and xenografts	[89]
*miR-9*	Overexpressed	CREB	*CREB,* *NF1*	Decreases proliferation and increases migration in glioma cells	[90]
*miR-29a*	Downregulated	-	*PDGFC, PDGFA*	Decreases proliferation, cell viability, migration, and invasion in glioma cells, and tumor growth in xenografts	[91]
*miR-95-3p*	Downregulated	-	*CELF2*	Decreases proliferation, cell viability, and invasion in glioma cells	[92]
*miR-124*	Downregulated	-	*SNAl2*	Decreases proliferation and invasion in glioma cells and tumor growth in xenografts	[93]
*miR-134*	Downregulated	-	*KRAS,* *STAT5B*	Decreases proliferation and cell viability in glioma and glioma stem cells and tumor growth in xenografts	[94]
*miR-145*	Downregulated	-	*-*	Decreases migration and invasion in glioma cells	[95]
*miR-148a*	Overexpressed	-	*MIG6,* *BIM*	Increases proliferation, cell viability, migration, and invasion in glioma cells, and tumor growth in xenografts	[96]
*miR-150*	Overexpressed	-	*VHL*	Increases glucose uptake, lactate secretion, and proliferation in glioma cells, and tumor growth in xenografts	[88]
*miR-181b*	Downregulated	-	*SP1*	Decreases glucose uptake and proliferation in glioma cells and tumor growth in xenografts	[97]
*miR-181d*	Downregulated	-	*KRAS,* *Bcl-2*	Decreases proliferation and cell viability in glioma cells and tumor growth in xenografts	[98]
*miR-203*	Downregulated	-	-	-	[99]
*miR-338-3p*	Downregulated	*circSMO742* &SMO	*-*	Decreases proliferation, cell viability, migration, and invasion in glioma cells	[100]
*miR-351*	Overexpressed	*-*	*NAIF1*	Increases cell viability, migration, and invasion in glioma cells	[101]
*miR-378e*	Downregulated	*circNFIX*	*RPN2*	Decreases glucose uptake, lactate secretion, cell viability, migration, and invasion in glioma cells	[102]
*miR-423-5p*	Overexpressed	-	*ING-4*	Increases proliferation, invasion, angiogenesis, and temozolomide resistance in glioma cells and tumor growth and invasion in xenografts	[103]
*miR-432-5p*	Downregulated	-	*RAB10*	Decreases glucose uptake, lactate secretion, invasion, and proliferation in glioma cells	[104]
*miR-451*	Downregulated	-	*-*	Increases cell viability and decreases invasion in glioma cells	[86]
*miR-451*	Downregulated	-	*CAB39*	Decreases proliferation, invasion, and migration in glioma cells and tumor growth in xenografts	[87]
*miR-451*	Downregulated in low glucose level glioma cells	-	-	Decreases migration in glioma cells andinvasion in xenografts, increases sensitivity to temozolomide treatment in glioma cells	[84]
*miR-451*	Downregulated	-	-	Decreases proliferation, cell viability, and invasion in glioma cells	[105]
*miR-451*	Downregulated	lncRNA *LSINCT5*	*CAB39*	Decreases glycolysis, cell viability, invasion, and migration in glioma cells	[85,106]
*miR-495*	Downregulated	*-*	*GLUT1*	Decreases glucose uptake and lactate secretion in glioma cells	[82]
*miR-663*	Downregulated	*-*	*PIK3CD*	Decreases proliferation and invasion in glioma cells	[107]

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
