# Peer review of "The Acidic Brain—Glycolytic Switch in the Microenvironment of Malignant Glioma"

_ijms, 2021, doi:10.3390/ijms22115518_

Round 1

Reviewer 1 Report

In the current review, the authors provide a detailed literature update of studies on the glycolytic microenvironment shift occurring in glioma development. The article is easy to read and well structured, however, it would be interesting and useful to  include a short section concerning the possible role played by microRNA/ncRNA in this metabolic switch.

Author Response

Response to reviewer 1 comments

We would like to thank the reviewer for his/her constructive criticism of the original manuscript. In this revision, we have attempted to address all the expressed concerns by including new literature and tables, and editing the text (changes are marked). These adaptations provide strong support and new insights to our review. They have further increased mechanistic insights into metabolic changes and tumor malignancy with respect to dysregulation of cellular micro- and non-coding RNAs – points that were raised by the reviewer. As a result, the revised manuscript does a better job of putting this topic into a broader biological context.

Point by point response:

In the current review, the authors provide a detailed literature update of studies on the glycolytic microenvironment shift occurring in glioma development. The article is easy to read and well structured, however, it would be interesting and useful to include a short section concerning the possible role played by microRNA/ncRNA in this metabolic switch.

Response: We thank the reviewer for appreciating our work and taking the time and effort to review our manuscript. We included a new section discussing the role of microRNAs in regulating glycolytic metabolism in glioma and their role for tumor malignancy. The following text has been included in the previous version of the manuscript: page 10 line 385 – page 12 line 412.

Reviewer 2 Report

In this review, the authors described the metabolic symbiosis model via lactate shuttling in glioma cells and highlighted its impacts of the glycolytic switch to glioma malignancy. The authors further identified glycolytic metabolism in the context of IDH mutation. This indicates that glycolytic switch of tumor cells might result in a more aggressive and malignant phenotype with enhanced cell proliferation, survival, migration and invasiveness, and angiogenesis. At last, the authors suggested that targeting glycolytic metabolisms, such as MCT inhibitors and glycolytic inhibitors could serve as potential therapeutic approaches for glioma patients. Overall, I think this manuscript is very informative, well-referenced, and well written. The manuscript is in good shape with some issues, the authors should improve the manuscript before it could be published.

Major:

  1. In line 269-270, Ref 62 indicated that: a) it is the phosphorylation of cortactin that increased the local pH, not vice versa; b) Increased pH led to the cofilin activation and invadopodium maturation, does this mean a more “alkaline” environment promote tumor cells migration and invasion? Seems contradictory with the conclusion in this manuscript. What are the authors' comments on these?
  2. The authors spent most of their efforts reviewing the Lactic acid metabolism as well as the structure and functions of MCT, which was very informative and in detail. In contrast, other parts of the manuscript seem to be superficial, such as the correlation between IDH mutation and HIF1-a. The length and the information across different sections are “imbalanced”. It is recommended that the author expand sections 3 and 4. For example, what is a correlation between IDH mutation and HIF1-a, there are some controversial studies (for example, PMID: 19359588 and PMID: 22343896), which the authors might want to expand the discussion in their manuscript. There are also a couple of review articles commenting on this topic (PMID: 32291392 and PMID: 29805076).
  3. The authors indicated that IDH mutant gliomas have higher basal respiration rates compared with their wild-type counterparts. This statement cannot be supported sufficiently by the references provided. Ref 87 was done in NHA cells, which is not a glioma cell line; Ref 88 used HCT116 cells which is not a glioma cell line either. In contrast, there are studies that reported that IDH mutant glioma cells exhibited significantly decreased oxidative metabolism, with a decrease in both baselines and the maximum capacity of oxygen consumption rate (PMID:28202508). The authors should investigate and discuss these controversial findings before they made such statements.
  4. Glycolytic metabolism and acid microenvironment may impact cancer immunity in many aspects including tumor immune surveillance, which eventually contributes to immune escape and cancer progression. Also, the low pH will also alter the therapeutic effects of immunotherapy (For example, PMID: 24091329, PMID: 17255361). The authors did mention that secretes branched-chain ketoacids via MCT1 affect TAMs and play a role in tumor immune suppression (line 298-286) but didn’t describe in detail. A more comprehensive discussion on the impact of tumor immunity and immunotherapy will be interesting.

Minor:

  1. It would be better if you include the full name of pO2 (partial pressure of oxygen) the first time it appears.
  2. Since this manuscript has many abbreviations, a list of abbreviations would be helpful.
  3. In Figure 3, the authors used both solid arrow and dash arrow. For example, in glycolytic glioma cells, the uptake of Glc was using a solid line, whereas, in oxidative glioma cells, the uptake of Glc was using a dashed line. What is the difference between them?
  4. In Figure 3, is the Glc uptake by glycolytic glioma cells derived from the oxidative glioma cells? Also, what is the gradient for lactate?
  5. The name of Ref 85 is not even completed. Please double-check for similar mistakes.

Author Response

Response to reviewer 2 comments

We would like to thank the reviewer for his/her constructive criticism of the original manuscript. In this revision, we have attempted to address all the expressed concerns by including new literature, adding illustrations and tables, and editing the text (changes are marked). These adaptations provide strong support and new insights to our review. They have further increased mechanistic insights into metabolic changes and tumor malignancy with respect to dysregulation of cellular nutrition and immunogenicity – points that were raised by the reviewer. As a result, the revised manuscript does a better job of putting this topic into a broader biological context.

Point by point response:

In this review, the authors described the metabolic symbiosis model via lactate shuttling in glioma cells and highlighted its impacts of the glycolytic switch to glioma malignancy. The authors further identified glycolytic metabolism in the context of IDH mutation. This indicates that glycolytic switch of tumor cells might result in a more aggressive and malignant phenotype with enhanced cell proliferation, survival, migration and invasiveness, and angiogenesis. At last, the authors suggested that targeting glycolytic metabolisms, such as MCT inhibitors and glycolytic inhibitors could serve as potential therapeutic approaches for glioma patients. Overall, I think this manuscript is very informative, well-referenced, and well written. The manuscript is in good shape with some issues, the authors should improve the manuscript before it could be published.

Response: We thank the reviewer for appreciating our work and taking the time and effort to review our manuscript.

Major:

  1. In line 269-270, Ref 62 indicated that: a) it is the phosphorylation of cortactin that increased the local pH, not vice versa; b) Increased pH led to the cofilin activation and invadopodium maturation, does this mean a more “alkaline” environment promote tumor cells migration and invasion? Seems contradictory with the conclusion in this manuscript. What are the authors' comments on these?

Response: We thank the reviewer for pointing this out. In the paper, the authors refer to the intracellular pH. Increased intracellular pH leads to invadopodia formation. When protons are exported via MCT4 together with lactate, intracellular pH increases and extracellular pH decreases. Therefore, the finding is consistent with the association of enhanced migration and decreased extracellular pH. To clarify this point, we edited the manuscript text (page 8 lines 321-322).  

  1. The authors spent most of their efforts reviewing the Lactic acid metabolism as well as the structure and functions of MCT, which was very informative and in detail. In contrast, other parts of the manuscript seem to be superficial, such as the correlation between IDH mutation and HIF1-a. The length and the information across different sections are “imbalanced”. It is recommended that the author expand sections 3 and 4. For example, what is a correlation between IDH mutation and HIF1-a, there are some controversial studies (for example, PMID: 19359588 and PMID: 22343896), which the authors might want to expand the discussion in their manuscript. There are also a couple of review articles commenting on this topic (PMID: 32291392 and PMID: 29805076).

Response: This is a good point. We have taken the reviewer’s concern into account and extensively expanded the respective paragraphs by discussing more literature and adding a new figure (Figure 4, page 13) for clarification. We discuss controversial results describing the relationship between IDH mutation status and HIF1-α (page 14 line 414 – page 14 line 480) and elaborated the subsections about diagnostics and therapy (see comments below), as the reviewer requested.

  1. The authors indicated that IDH mutant gliomas have higher basal respiration rates compared with their wild-type counterparts. This statement cannot be supported sufficiently by the references provided. Ref 87 was done in NHA cells, which is not a glioma cell line; Ref 88 used HCT116 cells which is not a glioma cell line either. In contrast, there are studies that reported that IDH mutant glioma cells exhibited significantly decreased oxidative metabolism, with a decrease in both baselines and the maximum capacity of oxygen consumption rate (PMID:28202508). The authors should investigate and discuss these controversial findings before they made such statements.

Response: We agree with the reviewer that the respective study (Ref 87, now Ref 123) was not conducted in glioma cell lines. However, the authors of this study applied immortalized human astrocytes expressing mutant or wildtype IDH1 (NHAIDHmut and NHAIDHwt). Since IDH mutation is one of the first events occurring in evolving glioma cells, in our opinion this model reflects these aspects of glioma evolution very well. In Ref. 88 (now Ref 119), the authors analyzed 112 IDH1WT versus 399 IDH1MUT low-grade glioma and 157 IDH1WT versus 9 IDH1MUT glioblastoma samples from patients. In both glioma types, IDH1WT was associated with high expression levels of genes encoding enzymes that are involved in glycolysis and acetate anaplerosis, whereas IDH1MUT gliomas overexpressed genes encoding enzymes involved in the oxidative TCA cycle. To better clarify this issue, we modified the statement about the basal respiration rates. However, the finding in the patient samples supports the glycolytic vs. oxidative phenotype in IDH wildtype vs. mutant gliomas. The text was edited accordingly (page 13 lines 438 – 444).

  1. Glycolytic metabolism and acid microenvironment may impact cancer immunity in many aspects including tumor immune surveillance, which eventually contributes to immune escape and cancer progression. Also, the low pH will also alter the therapeutic effects of immunotherapy (For example, PMID: 24091329, PMID: 17255361). The authors did mention that secretes branched chain ketoacids via MCT1 affect TAMs and play a role in tumor immune suppression (line 298-286) but didn’t describe in detail. A more comprehensive discussion on the impact of tumor immunity and immunotherapy will be interesting.

Response: We thank the reviewer for bringing this up. We have discussed glioma immunity and immunotherapy in greater depth in a separate subsection (page 15 line 556 – page 17 line 658), as the reviewer indicated.

Minor:

  1. It would be better if you include the full name of pO2 (partial pressure of oxygen) the first time it appears.

Response: We have included the full name in the text: page 2 line 71

  1. Since this manuscript has many abbreviations, a list of abbreviations would be helpful.

Response: A list of abbreviations has been added to the manuscript (page 18 line 689- page 20 line 809).

  1. In Figure 3, the authors used both solid arrow and dash arrow. For example, in glycolytic glioma cells, the uptake of Glc was using a solid line, whereas, in oxidative glioma cells, the uptake of Glc was using a dashed line. What is the difference between them?

Response: We apologize for this confusion. There was no different meaning intended. We edited the Figure 3 (page 8) by only using solid lines in the updated version of the manuscript.

  1. In Figure 3, is the Glc uptake by glycolytic glioma cells derived from the oxidative glioma cells? Also, what is the gradient for lactate?

Response: We thank the reviewer for pointing this out. Glucose taken by glycolytic glioma cells is derived from blood vessels. We have edited the figure legend (page 8 lines 304-305) to prevent further confusion. Furthermore, we have added the lactate gradient to the figure.

  1. The name of Ref 85 is not even completed. Please double-check for similar mistakes.

Response: This is a good point. All new and previous references have been checked for completion.
